# Changes in susceptibility of *Plasmodium falciparum* to antimalarial drugs in Uganda over time: 2019–2024

Martin Okitwi [1,5], Stephen Orena[1,5], Patrick K. Tumwebaze [1], Thomas Katairo[1], Yoweri Taremwa[1], Oswald Byaruhanga[1], Stephen Tukwasibwe[1], Samuel L. Nsobya[1], Jennifer Legac[2], Jeffrey A. Bailey [3], Roland A. Cooper[4], Melissa D. Conrad [2,6] ✉ & Philip J. Rosenthal [2,6] ✉

The treatment and control of malaria in Africa is challenged by drug resistance. We characterized ex vivo susceptibilities to nine drugs of isolates collected from individuals presenting with uncomplicated falciparum malaria in eastern (2019-2024) and northern (2021-2024) Uganda and performed deep sequencing, with analysis of 80 *Plasmodium falciparum* genes, to evaluate associations between susceptibilities and potential resistance markers for samples studied since 2016. For 1114 evaluated isolates, median half-maximal inhibitory concentrations (IC$_{50}$s) were low-nanomolar for chloroquine, monodesethylamodiaquine, piperaquine, pyronaridine, lumefantrine, mefloquine, and DHA, but higher for quinine and pyrimethamine. Over time, susceptibilities improved for chloroquine, decreased for lumefantrine, mefloquine, and DHA, and were unchanged for other drugs. Changes in prevalences of known markers of altered drug susceptibility followed the same patterns. Genotypes associated with drug susceptibility were those previously identified for aminoquinolines and pyrimethamine. For lumefantrine, susceptibility was decreased with wild-type PfCRT K76T or PfMDR1 N86Y, mutant PfK13 C469Y or A675V, mutant PfCARL D611N, and other polymorphisms. For DHA, susceptibility was decreased with the PfK13 C469Y or A675V and PfMDR1 Y500N mutations. Decreasing activities of lumefantrine and DHA suggest potential loss of efficacies of leading regimens, although the clinical consequences of these changes are, to date, uncertain.

After good progress early this century, the control of malaria in Africa, the region responsible for ~95% of malaria morbidity and mortality, has stalled[1]. A challenge to malaria control and eventual elimination is increasing resistance of malaria parasites, in particular *Plasmodium falciparum*, to available drugs[2]. Of particular concern is potential resistance to artemisinin-based combination therapies (ACTs), including artemether-lumefantrine and artesunate-amodiaquine, the most widely used therapies for uncomplicated malaria in Africa, and the alternative ACTs dihydroartemisinin (DHA)-piperaquine, artesunate-pyronaridine, and artesunate-mefloquine[3]. In addition,

[1]Infectious Diseases Research Collaboration, Kampala, Uganda. [2]Department of Medicine, University of California, San Francisco, CA, USA. [3]Brown University, Providence, RI, USA. [4]Department of Natural Sciences and Mathematics, Dominican University of California, San Rafael, CA, USA. [5]These authors contributed equally: Martin Okitwi, Stephen Orena. [6]These authors jointly supervised this work: Melissa D. Conrad, Philip J. Rosenthal.
✉e-mail: melissa.conrad@ucsf.edu; philip.rosenthal@ucsf.edu

sulfadoxine-pyrimethamine (SP) has an important role in malaria control, including intermittent preventive therapy in pregnant women, perennial malaria chemoprevention in infants, and, in combination with amodiaquine, seasonal malaria chemoprevention in children[4]. Increasing resistance to artemisinins, ACT partner drugs, and SP threatens effective treatment and prevention of malaria across Africa.

Three categories of antimalarial drug resistance are highly relevant for Africa[5,6]. First, resistance to chloroquine and amodiaquine is mediated principally by the PfCRT K76T and PfMDR1 N86Y mutations, which alter drug transport[7]. The prevalence of these mutations was previously high across Africa, but it has decreased greatly in many areas[5]. An opposite effect is seen with lumefantrine and mefloquine, with the PfCRT K76T and PfMDR1 N86Y wild-type alleles and *pfmdr1* amplification associated with decreased susceptibility[8]. Different mutations in PfCRT and amplification of plasmepsin genes have been associated with resistance to piperaquine, a related aminoquinoline, but to date only in southeast Asia[6] and South America[9]. Second, partial resistance to artemisinins (ART-R) became well established in the Greater Mekong Subregion of southeast Asia early this century[10]. ART-R manifests as delayed parasite clearance after therapy with artemisinins and enhanced parasite survival after in vitro exposure to DHA, and is mediated principally by any of ~20 mutations (with generally only one of these mutations in an individual strain) in the *P. falciparum* kelch (PfK13) protein propeller domain[11]. Multiple PfK13 mutations previously validated as mediators of ART-R have emerged in Africa, including R561H in Rwanda[12,13], western Tanzania[14], and southwestern Uganda[15]; C469Y and A675V in northern Uganda[15,16]; and R622I in Eritrea and Ethiopia[17,18]. However, these mutations have not been clearly linked to decreased clinical efficacy of ACTs in Africa, and associations between PfK13 mutations and resistance phenotypes do not appear to be as straightforward in Africa as in Southeast Asia[11]. Third, resistance to SP, mediated principally by mutations in the target enzymes dihydrofolate reductase (PfDHFR N51I, C59R, S108N) and dihydropteroate synthase (PfDHPS A437G, K540E), is common in much of Africa, with two additional mutations, PfDHFR I164L and PfDHPS A581G, which mediate high-level SP resistance[19], spreading across parts of east Africa[5,20,21].

In Uganda, serial surveillance has shown loss of the PfCRT and PfMDR1 mutations that mediate resistance to chloroquine and amodiaquine, emergence of five validated or candidate PfK13 ART-R mutations in different parts of the country, and increasing prevalence of the PfDHFR I164L and PfDHPS A581G SP-resistance mutations[15,21]. Prior ex vivo studies in Tororo, in eastern Uganda, showed generally good activities of studied antimalarials, except for the PfDHFR inhibitor pyrimethamine[22,23]. However, modest decreases in susceptibility to DHA and lumefantrine, which may challenge the efficacy of artemether-lumefantrine, the first-line malaria therapy, were recently seen in isolates from northern Uganda[24]. Most clinical trials have shown excellent efficacies of leading ACTs in Africa[5,25,26] but there have been exceptions (at sites without known ART-R at the times of the studies) with artemether-lumefantrine genotype-corrected treatment efficacies <90%[27], including a recent trial in Uganda[28], although genotyping to assign outcomes is challenging in such high transmission regions.

Considering the dynamic nature of antimalarial drug sensitivity in Africa, it is important to maintain surveillance for emerging drug resistance. For this reason, we have regularly assessed ex vivo drug susceptibilities and genotypes of *P. falciparum* causing malaria in eastern Uganda since 2010 and in northern Uganda since 2021. We report here the results of surveillance conducted over the last five years and associations between drug susceptibility phenotypes and parasite genotypes.

## Results
### Study samples and participants
Of 1297 *P. falciparum* isolates collected since July, 2019, 724/828 assessed in Tororo, in eastern Uganda, and 390/469 assessed in

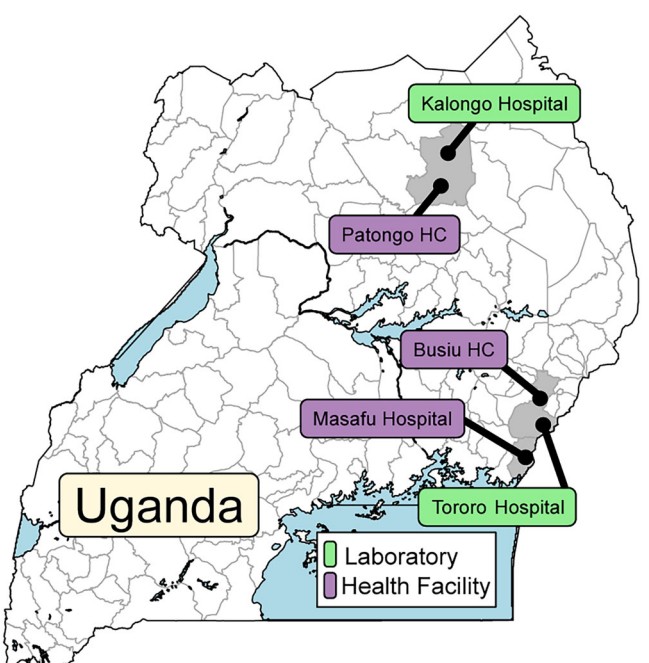

**Fig. 1 | Study sites.** Samples were collected at the indicated health facilities and at clinics adjacent to the two indicated laboratories.

Kalongo, in northern Uganda, were successfully evaluated for ex vivo drug susceptibilities (Fig. 1). Baseline characteristics of participants were similar over time, although parasitemias were lower and participant ages higher in northern Uganda (Table 1).

### Ex vivo drug susceptibilities
We measured ex vivo susceptibilities of all isolates to nine standard antimalarials. For successful assays, the mean Z factor was 0.75 (SD ± 0.26), indicating robust assays. Median $IC_{50}$ values were at low nanomolar levels for chloroquine (12.6 nM), MDAQ (7.8 nM), piperaquine (5.4 nM), DHA (2.9 nM), lumefantrine (11.3), mefloquine (15.2 nM), and pyronaridine (1.5 nM), consistent with potent activity, and higher for quinine (115 nM), which is typically less potent than the other studied compounds, and pyrimethamine (35,100 nM), against which resistance is well-established (Table 2; Fig. 2). From 2016 to 2024, marked decreases in susceptibilities (Mann-Kendall Tau ≥±0.2) were seen for DHA, lumefantrine, and mefloquine (Supplementary Table 1). The more marked changes in susceptibilities occurred in eastern Uganda, likely with decreases in susceptibilities to key drugs in northern Uganda before initiation of studies in that region, as suggested by our earlier comparison of the sites (Supplementary Figs. 1 and 2)[24]. Susceptibilities of Dd2 and 3D7 laboratory reference strains yielded $IC_{50}$ values similar to those reported previously (Supplementary Table 2). Comparing median values for five drugs studied in both 2010–2013[22] and the current study, susceptibilities increased for chloroquine ($IC_{50}$ 248 to 12.6 nM), MDAQ (76.9 to 7.8 nM), and piperaquine (19.1 to 5.4 nM) and decreased for DHA (1.4 to 2.9 nM) and lumefantrine (3.0 to 11.3 nM; $p < 0.001$ for all comparisons).

RSAs were performed on a subset of isolates, including 126 from eastern and 314 from northern Uganda. Consistent with other recent results from Uganda[24], but in contrast to results from before 2020 in Uganda[23,29] and from southeast Asia[30], many isolates had survival above previously-established ART-R cut-offs (Fig. 2). Overall, 78.2% of isolates had 72 h survival >1%, and 35.8% survival >5% of control values. RSA survival increased over time, but with the significance criterion of Mann-Kendall Tau ≥±0.2 the increase was significant only in eastern Uganda.

**Table 1 | Characteristics of study participants and samples studied by ex vivo analysis**

| Year | All sites | | | | | Eastern Uganda | | | | | Northern Uganda | | | | |
|---|---|---|---|---|---|---|---|---|---|---|---|---|---|---|---|
| | Samples studied | Parasitemia (median, %, (IQR)) | Male (%) | Age (mean, y (SD)) | <5 y age (%) | Samples studied | Parasitemia (median, %, (IQR)) | Male (%) | Age (mean, y (SD)) | <5 y age (%) | Samples studied | Parasitemia (median, %, (IQR)) | Male (%) | Age (mean, y (SD)) | <5 y age (%) |
| 2019 | 76 | 3.4 (2.0–5.6) | 34.2 | 5.4 (4.9) | 54.0 | 76 | 3.4 (2.0–5.6) | 34.2 | 5.4 (4.9) | 54.0 | 0 | - | - | - | - |
| 2020 | 97 | 4.0 (2.5–6.1) | 39.6 | 5.2 (4.3) | 60.4 | 97 | 4.0 (2.5–6.1) | 39.6 | 5.2 (4.3) | 60.4 | 0 | - | - | - | - |
| 2021 | 210 | 2.4 (1.2–4.2) | 40.5 | 7.6 (6.7) | 42.4 | 154 | 3.1 (1.8–4.9) | 41.6 | 6.2 (5.2) | 48.1 | 56 | 1.0 (0.5–1.6) | 37.5 | 11.3 (8.7) | 26.8 |
| 2022 | 262 | 1.8 (0.9–3.3) | 45.4 | 8.3 (7.1) | 36.9 | 162 | 2.4 (1.6–4.0) | 48.2 | 5.8 (4.5) | 50.7 | 100 | 0.9 (0.5–1.7) | 41.0 | 12.0 (8.7) | 16.0 |
| 2023 | 313 | 1.7 (0.9–3.3) | 40.9 | 7.5 (6.8) | 42.2 | 171 | 2.5 (1.2–4.5) | 46.4 | 6.0 (7.1) | 57.8 | 142 | 1.0 (0.5–1.8) | 34.5 | 9.2 (6.1) | 23.9 |
| 2024 | 155 | 1.5 (0.7–2.9) | 50.3 | 7.2 (7.7) | 48.1 | 64 | 2.1 (1.5–3.3) | 48.4 | 6.3 (5.7) | 57.1 | 91 | 0.9 (0.43–2.0) | 51.7 | 7.8 (8.9) | 41.8 |
| Total | 1113 | 2.0 (1.0–3.9) | 42.6 | 7.3 (6.8) | 44.3 | 724 | 2.7 (1.6–4.9) | 43.7 | 5.9 (5.5) | 54.0 | 389 | 1.0 (0.5–1.8) | 40.6 | 9.9 (8.0) | 26.5 |

## Genotypes

Of the 1114 isolates with ex vivo results, we characterised sequences of 1070 for known markers of altered drug susceptibility. Prevalences of the PfCRT K76T and PfMDR1 N86Y mutations, which are associated with resistance to chloroquine and amodiaquine[7], have been decreasing[21], and were very low in recent years in our studied isolates (Fig. 3; Supplementary Table 3). Prevalences of two other common PfMDR1 mutations were similar to those reported previously[21], with stable prevalence of the Y184F mutation, which has generally not been associated with drug susceptibility, but may impact on parasite fitness[31], and low and decreasing prevalence of the D1246Y mutation. Mutations associated with aminoquinoline resistance in Southeast Asia (PfCRT H97Y, F145I, M343L, G353V[32]) or South America (PfCRT C350R[9]) were not detected in any isolates. Definitive increased copy number of *pfmdr1*[33] or *plasmepsin* 2/3[9,34], which has been associated with decreased susceptibility to lumefantrine and mefloquine or piperaquine, respectively, was not observed, with the vast majority of copy numbers measured at ≤1.5, and copy number of 1.6 seen for 1/661 isolates for *pfmdr1* and 4/445 isolates for *plasmepsin* 2 (Supplementary Table 4). Prevalences of five mutations associated with resistance to SP (PfDHFR N51I, C59R, S108N; PfDHPS A437G, K540E) were very high, as has been the case in Uganda for at least two decades[35], and two additional mutations associated with higher level resistance (PfDHFR I164L, PfDHPS A581G[19]), and seen in recent years at increasing prevalence in western Uganda[21], had modest prevalence in both eastern and northern Uganda (Fig. 3; Supplementary Table 3). The PfK13 C469Y and A675V mutations were first identified in southeast Asia and more recently validated as markers of ART-R in northern Uganda[11,16,24]. These mutations were at moderate prevalence in northern Uganda at the time of our first collections in 2021 (prevalence 30.5% for C469Y and 8.5% for A675V), with stable prevalence since that time. In eastern Uganda, the mutations were at very low prevalence until 2021, with increasing prevalence since that time. Other PfK13 validated or candidate ART-R mutations that have been seen elsewhere in eastern Africa (P441L, C469F, R561H, R622I) were not seen.

## Genotype-phenotype associations

We searched for associations between genotypes identified by sequencing of 80 candidate genes and drug susceptibility phenotypes, considering available data from 2016 to 2024, including older results published previously[24], and we included strict criteria for significant associations. Of greatest interest were results for DHA and lumefantrine. Considering PfK13 mutations previously associated with ART-R, the C469Y and A675V mutations were associated with decreased activity (based on $IC_{50}$s) for DHA and lumefantrine (Fig. 4, Supplementary Tables 5 and 6). Interestingly, these mutations were not associated with RSA results. As described previously[8], the PfMDR1 N86Y wild-type allele was associated with decreased susceptibility to lumefantrine and mefloquine and the PfCRT K76T wild-type allele with decreased susceptibility to lumefantrine, although analyses were limited by low prevalence of mutant genotypes. Multiple other polymorphisms were associated with susceptibilities to DHA and lumefantrine (Supplementary Tables 5 and 6). Strong associations included, for lumefantrine, PfCARL D611N ($IC_{50}$ 7.5 nM for wild type, 20.2 nM for mixed, and 44.3 nM for mutant), with increased prevalence over time, and, for DHA, PfMDR1 Y500N ($IC_{50}$ 1.9 nM for wild type, 2.6 nM for mixed, and 5.4 nM for mutant; Fig. 4, Supplementary Tables 5 and 6).

To consider the interplay between PfK13 mutations and candidate resistance markers, we assessed shared prevalence. The few PfMDR1 N86Y and PfCRT K76T mutations identified were seen exclusively with PfK13 wild-type sequences. Considering only pure mutant and pure wild-type genotypes to avoid haplotype assumptions, PfCARL D611N was seen with PfK13 wild-type, C469Y mutant, and A675V mutant

## Table 2 | Summary of drug susceptibility data (2019–2024)

| Drug | IC$_{50}$ (nM) | | | | | | | | |
| --- | --- | --- | --- | --- | --- | --- | --- | --- | --- |
| | All sites | | | Eastern Uganda | | | Northern Uganda | | |
| | N | Median | IQR | N | Median | IQR | N | Median | IQR |
| Chloroquine | 1060 | 12.6 | 9.2–18.0 | 707 | 13.2 | 9.7–19.0 | 353 | 11.3 | 8.7–16.4 |
| MDAQ | 1062 | 7.8 | 4.9–10.4 | 708 | 7.6 | 4.8–10.1 | 354 | 8.2 | 5.2–10.8 |
| Piperaquine | 1062 | 5.4 | 3.6–8.2 | 708 | 5.5 | 3.6–8.3 | 354 | 5.2 | 3.5–7.8 |
| DHA | 1063 | 2.9 | 1.6–4.5 | 710 | 2.5 | 1.5–4.1 | 353 | 3.4 | 2.2–5.0 |
| Lumefantrine | 1060 | 11.3 | 6.1–19.5 | 706 | 8.7 | 5.2–17.2 | 354 | 15.3 | 10.4–21.7 |
| Mefloquine | 1058 | 15.2 | 9.6–24.8 | 709 | 14.9 | 9.4–25.2 | 349 | 15.8 | 9.9–23.2 |
| Pyronaridine | 1057 | 1.5 | 0.7–2.7 | 704 | 1.5 | 0.8–2.8 | 353 | 1.4 | 0.6–2.7 |
| Quinine | 770 | 115 | 75.1–173 | 625 | 117 | 76.8–178 | 145 | 106 | 72.1–153 |
| Pyrimethamine | 881 | 35,100 | 24,900–49,100 | 674 | 38,100 | 27,100–52,000 | 207 | 28,100 | 19,900–37,400 |

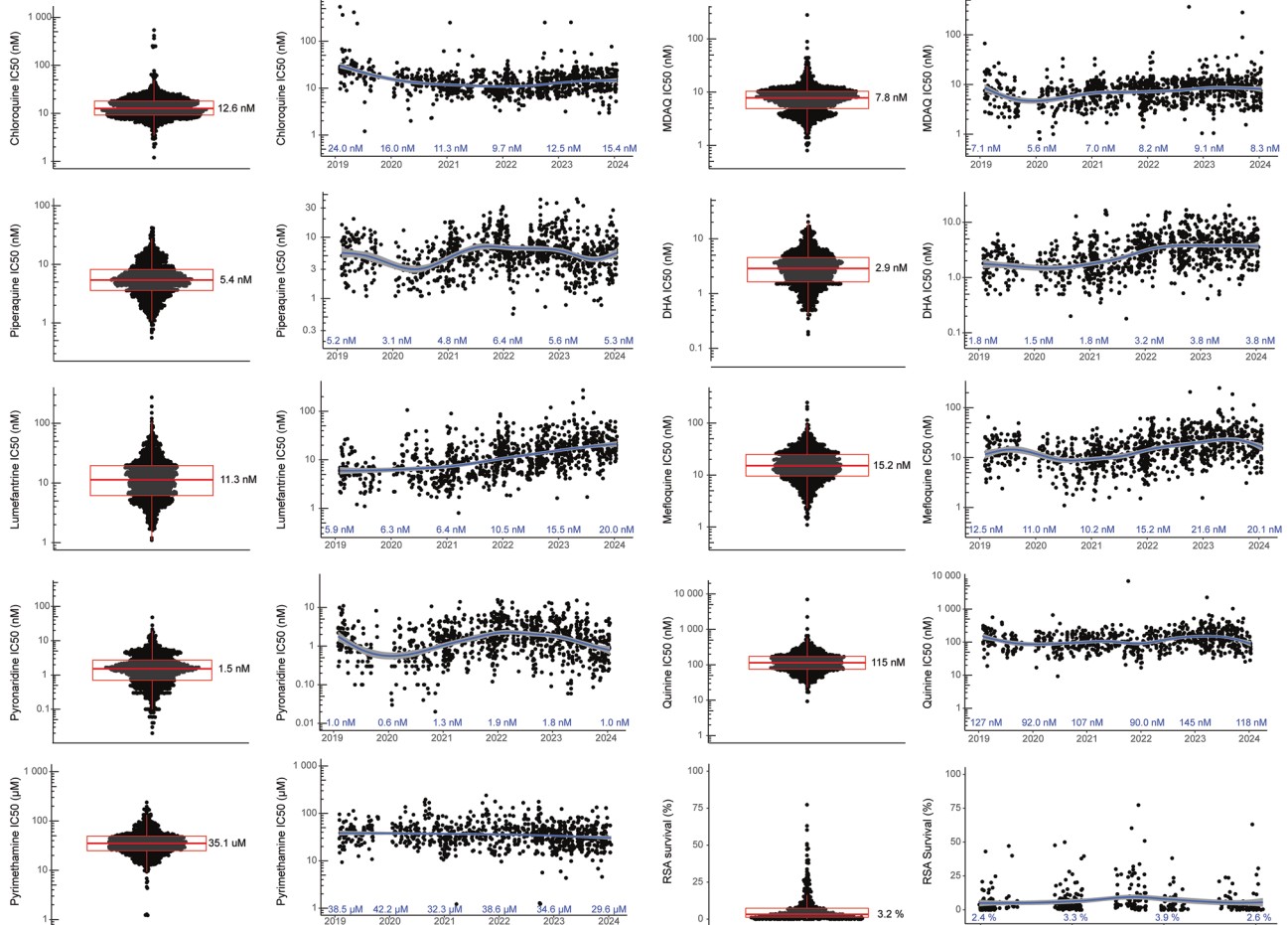

**Fig. 2 | Ex vivo drug susceptibilities in Uganda over time.** Paired plots present the distribution of susceptibilities for assayed isolates (left; median values shown) and results over time (right; median values shown for each year). Each dot represents an isolate; n for each drug and each year is provided in Supplementary Table 9. There were no technical replicates. Boxes show the first to third quartiles, and whiskers extend to the largest values no more than 1.5X the inter-quartile ranges. X-axis tick marks indicate June 1 of each year. The curves were generated using loess smoothing implemented by geom_smooth; the grey bands represent the 95% confidence intervals. Source data are provided as a Source Data file.

sequences, while PfMDR1 Y500N was seen with PfK13 wild-type and C469Y mutant sequences. Susceptibilities to lumefantrine were lowest in the presence of PfK13 C469Y and/or A675V mutations with either wild-type or mutant alleles at PfCARL D611N and for DHA were lowest in the presence of the PfK13 C469Y mutation with either wild-type or mutant alleles at PfMDR1 Y500N, although few samples were available for some comparisons (Supplementary Table 7). As with the PfK13

mutations, the PfCARL D611N and PfMDR1 Y500N mutations were not associated with RSA results. The mutations were also associated with decreased susceptibilities in the absence of PfK13 mutations.

For chloroquine and amodiaquine, consistent with earlier results, the PfCRT K76T mutation (Fig. 4, Supplementary Table 8) and other PfCRT mutations that usually form a haplotype, were associated with decreased activity. For pyrimethamine, susceptibilities were poor with

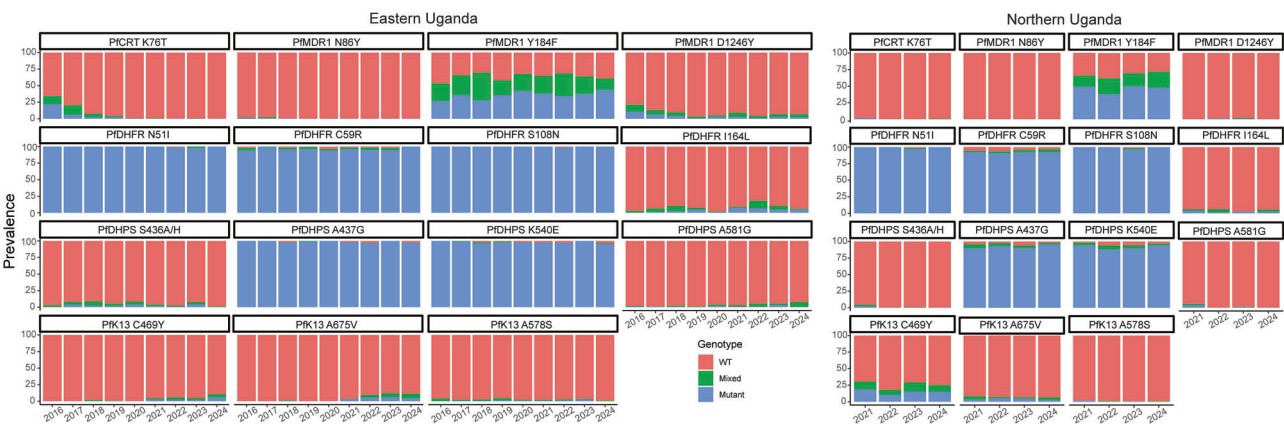

**Fig. 3 | Prevalence of genetic polymorphisms associated with altered drug susceptibility over time at sites in eastern and northern Uganda.** WT wild type. Source data are provided as a Source Data file.

**Fig. 4 | Associations between genotypes of interest and ex vivo drug susceptibility.** Drug susceptibility for wild-type (WT), mixed WT/mutant, and pure mutant (Mut) isolates are shown for selected drugs and polymorphisms of interest. There were no technical replicates. Centre bounds of boxes correspond to the medians, and minimal and maximal bounds correspond to 25 and 75th percentiles,

respectively. Whiskers extend to extreme values no further than 1.5x the IQR from the 25 or 75th percentiles. Benjamini-Hochberg corrected p-values for two-sided, pairwise-Wilcoxon tests are indicated. Additional associations are shown in Supplementary Tables 5-7. Source data are provided as a Source Data file.

the PfDHFR C59R mutation ($IC_{50}$ 37,800 nM) and even poorer with the I164L mutation (86,800 nM; Fig. 4, Supplementary Table 8). No significant associations were seen between studied genotypes and the ex vivo activities of piperaquine, pyronaridine, or quinine, or DHA activity based on the RSA.

## Discussion

In the face of changing drug susceptibilities, it is critical that we understand the antimalarial efficacies of key drugs in Africa. We have studied the ex vivo activities of antimalarial drugs against freshly isolated *P. falciparum* in Uganda since 2010. This report offers data

collected over the last five years. Commonly used drugs, including DHA, the active metabolite of all clinically relevant artemisinins, and all widely used ACT partner drugs (except SP, which is little used for treatment in Africa) remained active against *P. falciparum* in standard assays at low nM concentrations, a reassuring result. However, important changes in susceptibilities were seen. Specifically, susceptibility to chloroquine increased, suceptibilities to most other tested drugs remained stable, but susceptibilities to both DHA and the ACT partner drug lumefantrine decreased over time. Absolute changes in susceptibilities to DHA and lumefantrine were modest, and the clinical consequences of these changes are uncertain. Overall, our results offer reassurance that the activities of widely used antimalarials generally remain good, but also raise concern that the efficacies of leading ACTs may be decreasing.

Coincident with the emergence and spread of PfK13 mutations previously validated as markers of ART-R, susceptibility to DHA has decreased. However, significant decreases and significant correlations between PfK13 mutations and DHA susceptibility were seen only with the standard DHA growth inhibition assay and not with the RSA, which includes a 6 h incubation with DHA to mimic the short clinical exposure to artemisinins, and was a better indicator of ART-R than DHA IC$_{50}$ measurements in southeast Asia[30]. Factors contributing to the lack of significant correlation between PfK13 mutations and RSA survival may have included the polyclonal nature of most Ugandan isolates, technical differences between the ex vivo RSA (which does not include parasite synchronisation) and assays with synchronised culture adapted parasites, and our relatively small sample size for RSAs. Of note, enhanced parasite survival in the ex vivo RSA was more common than in studies of Ugandan isolates collected from 2015 to 2020 using the same assay[8,29], suggesting loss of DHA activity over time, and that mediators in addition to PfK13 mutations impact parasite clearance.

Lumefantrine, the partner drug for the most widely used ACT, has demonstrated potent activity and a strong barrier to resistance since the initiation of widespread use in Africa about two decades ago. However, lumefantrine susceptibility was lower in isolates collected in 2021 from northern, compared to eastern Uganda, and susceptibility was tightly correlated with prevalence of the PfK13 C469Y and A675V mutations, which at that time had high prevalence in northern, but not eastern Uganda[24]. Since that time, the prevalence of the C469Y and A675V mutations has increased, and lumefantrine susceptibility has decreased in eastern Uganda[15], and prevalence of these mutations was again tightly correlated with lumefantrine susceptibility. Although there is no direct evidence that decreased lumefantrine susceptibility is mediated by PfK13 mutations, it is possible that these PfK13 mutant parasites have acquired additional mutations that influence susceptibility to lumefantrine.

The clinical consequences of the observed changes in *P. falciparum* susceptibility to artemether-lumefantrine in Uganda remain uncertain. Most recent trials have demonstrated excellent treatment efficacies for artemether-lumefantrine[5,11,27]. However, studies in Angola[36], Burkina Faso[37], Democratic Republic of Congo[38], Tanzania[39], and Uganda[28] have shown genotype-corrected treatment efficacies for AL < 90% at certain sites. In all cases, the prevalence of ART-R-mediating PfK13 mutations was very low at the time of these studies, so the sub-optimal treatment efficacy was not due in a straightforward manner to ART-R. Further, these results must be interpreted with caution, as assignment of treatment outcomes using varied molecular genotyping methods is inexact, especially in areas of very high malaria transmission[40–42]. In addition, *P. falciparum* infections recently acquired in Africa and evaluated elsewhere have demonstrated decreased in vitro activity of lumefantrine, enhanced RSA survival, and multiple failures of therapy with artemether-lumefantrine both with[43,44] and without[44,45] known ART-R-mediating PfK13 mutations. Taken together, although our understanding of this area remains incomplete, the combination of evidence for decreasing activities of

both components of artemether-lumefantrine in Uganda in prior studies[23,24] and this report and of sub-optimal clinical efficacy for artemether-lumefantrine in multiple countries is concerning.

We assessed associations between polymorphisms in 80 genes known or suspected to be linked to antimalarial drug resistance and ex vivo drug susceptibility. We confirmed previously identified associations for chloroquine and MDAQ (decreased susceptibility with the PfCRT K76T mutation), lumefantrine and mefloquine (decreased susceptibility with the wild-type PfMDR1 N86Y allele), and pyrimethamine (stepwise decreased susceptibility with the PfDHFR C59R and I164L mutations). These results suggest improved activity of ASAQ, consistent with recent therapeutic efficacy studies, modest decreases in lumefantrine activity, and minimal activity for pyrimethamine as a component of SP.

As shown previously, the PfK13 C469Y and A675V mutations were associated with decreased susceptibility to both DHA and lumefantrine[24]. Among other studied polymorphisms, multiple associations were seen, and of interest were strong associations for the PfCARL D611N mutation for lumefantrine and the PfMDR1 Y500N mutation for DHA; whether these polymorphisms directly impact drug susceptibility is unknown. The variability in drug susceptibilities for isolates without known drug resistance-mediating mutations suggests that additional genetic polymorphisms impact on susceptibility. Ongoing whole-genome sequencing of select isolates may identify additional potential mediators of altered drug susceptibility.

Our new data offer insights into mediators of drug susceptibility and the consequences of changing *P. falciparum* genetics in Uganda over time. First, for chloroquine and amodiaquine, drug susceptibility remains strongly correlated with the primary resistance mediator, the PfCRT K76T mutation, but this mutation is now very uncommon in Uganda. Second, susceptibility to DHA and lumefantrine was significantly decreased in parasites with the PfK13 C469Y or A675V mutations, which now have combined prevalences approaching 50% in much of Uganda[15]. Third, lumefantrine susceptibility was also modestly but significantly decreased in parasites with the wild-type PfMDR1 N86Y sequence, which is now nearly universal in Uganda. Fourth, other polymorphisms associated with susceptibility to DHA and lumefantrine were identified, highlighting additional potential resistance mediators. Fifth, pyrimethamine susceptibility was poor for isolates containing three common PfDHFR mutations and even worse for those that also contained the I164L mutation, suggesting that, consistent with recent studies from Uganda[46], the antimalarial protective efficacy of SP will be poor. Taken together, our results indicate a number of polymorphisms with clear impacts on *P. falciparum* drug susceptibility and evidence that susceptibilities to the components of artemether-lumefantrine have decreased. Thus, continued performance of parasitological and genomic surveillance for evidence of antimalarial drug resistance and institution of policy changes to limit resistance selection and treatment failure are high priorities[47].

## Methods
### Isolates for study
We collected blood from patients diagnosed with uncomplicated malaria between July, 2019 and June, 2024 at three sites in eastern Uganda (Tororo District Hospital, Tororo District; Masafu General Hospital, Busia District; and Busiu Health Centre IV, Mbale District) and assessed drug susceptibilities at our laboratory in Tororo (Fig. 1). We added collection of samples from patients with uncomplicated malaria in Agago District, in northern Uganda, first with transport of samples from Patongo Health Centre IV to our Tororo laboratory from May to July, 2021 and April to July, 2022, and since January, 2023 with samples from Patongo Health Centre IV and Dr. Ambrosoli Memorial Hospital, Kalongo assessed in a new laboratory at the hospital. To increase sample size for genotype-phenotype association studies we extended our analyses to include 1668 samples collected since 2016, including

results for 554 samples which were published previously, all from eastern Uganda and collected from June, 2016 to July, 2019[23]. Patients reporting antimalarial treatment within the previous 30 days or infected with non-falciparum species were excluded. Written informed consent was obtained from adults and parents or guardians of children <18 years; children aged 8–17 years provided assent. Venous blood (2–5 ml) was collected in a heparin tube, after which participants were treated with artemether-lumefantrine, following national guidelines. The study was approved by the Makerere University School of Biomedical Sciences Research and Ethics Committee, the Uganda National Council for Science and Technology, and the University of California, San Francisco Committee on Human Research.

### Determination of ex vivo drug susceptibilities

Samples containing only *P. falciparum* and at least 0.2% parasitaemia by Giemsa-stained thin smear were analysed. Samples were centrifuged, buffy coats removed, and erythrocyte pellets washed 3X with wash medium (RPMI 1640 with 25 mM HEPES, 24 mM $NaHCO_3$, 10 µg/mL gentamicin) and resuspended in complete culture medium (RPMI 1640 with 25 mM HEPES, 24 mM $NaHCO_3$, 0.1 mM hypoxanthine, 10 µg/mL gentamicin, and 0.5% AlbuMAX II (Thermo Fisher Scientific)) to produce a haematocrit of 50%, and aliquots were spotted onto filter paper for molecular analysis.

Drug susceptibilities were assessed using a 72 h growth inhibition assay with SYBR Green detection[23]. Briefly, study compounds (chloroquine, monodesethylamodiaquine (MDAQ, the active metabolite of amodiaquine), piperaquine, pyronaridine, mefloquine, lumefantrine, DHA, quinine, and pyrimethamine), supplied by Medicines for Malaria Venture, were dissolved in dimethyl sulfoxide (water for chloroquine) as 10 mM stocks (50 mM for pyrimethamine) stored at −20 °C, and three-fold serial dilutions in complete medium were placed in 96-well microplates (50 µL per well), including drug-free and parasite-free controls. Cultures were diluted with uninfected erythrocytes to 200 µL per well at 0.2% parasitaemia and 2% haematocrit. Plates were maintained at 5% $CO_2$, 5% $O_2$, and 90% $N_2$ for 72 h at 37 °C in a humidified modular incubator. After 72 h, 100 µL resuspended culture per well was transferred to black 96-well plates containing 100 µL SYBR Green lysis buffer (20 mM Tris, 5 mM EDTA, 0.008% saponin, 0.08% Triton X-100, and 0.2 µL/mL SYBR Green I) per well and mixed, plates were incubated for 1 h in the dark at room temperature, and fluorescence (485 nm excitation and 530 nm emission) was measured. $IC_{50}$ values were derived from plots of fluorescence intensity vs. log drug concentration and fit to non-linear curves using a four-parameter Hill equation in Prism. Z factors to assess well-to-well variability and signal-to-noise ratios were calculated as previously described[23]. When steep slopes resulted in poor curve fit, slopes were fixed to a constant value of −6. For results with incomplete curves at low drug concentrations but at least 50% of the curve present, the upper plateau was constrained to that of drug-free wells on the same plate. Control *P. falciparum* Dd2 (MRA-156) and 3D7 (MRA-102) strains from MR4/BEI Resources were maintained in culture, synchronised with a magnetic column, and assayed, beginning at the ring-stage, approximately monthly. The ex-vivo ring-stage survival assay, which entails comparing parasitaemias 66 h after a 6 h incubation with 700 nM DHA with that of untreated controls, was performed as previously described[23].

### Genotyping

For genotyping, parasite DNA was extracted from filter paper blood spots using Chelex-100 and analysed by molecular inversion probe (MIP) capture and deep sequencing, using a MIP panel with probes targeting 80 genes of interest, as previously described[21,24]. To resolve ambiguities, some PfK13 sequences were additionally analysed by dideoxy sequencing, as previously described[8]. Sequencing reads are available in the National Center for Biotechnology Information Archive (BioProject PRJNA850445). Raw sequencing data were analysed using MIPTools. Copy numbers were estimated based on sample and probe normalised depth of coverage from 31 unique probes for *pfmdr1* and 21 probes for *plasmepsin 2/3*; copy number >1.7 was considered multiple. The Dd2 strain, which has amplified *pfmdr1* and is single copy for *plasmepsin 2/3*, and the G8 clone of Cambodian isolate KH001_053, with one copy of *pfmdr1* and two copies of *plasmepsin 2/3*, were included as controls.

### Statistical analysis

All statistical tests were done in R Studio (version 2024.09.1 + 394). Baseline characteristics of participants and isolates were computed as prevalences, medians with interquartile ranges (IQR), or means with standard deviations. Summary statistics for ex vivo susceptibilities were median $IC_{50}$ with IQR. To assess well-to-well variability and signal-to-noise ratios in the fluorescence readout of ex-vivo assays, Z factors were calculated as: $Z = 1 − ([3 × SD_{infected\ drug\ free} + 3 × SD_{uninfected}] / [mean_{infected\ drug\ free} − mean_{uninfected}])$[48]. The Mann-Kendall non-parametric test was used to detect monotonic trends (change over time in a consistent positive or negative direction) in drug susceptibilities with the Kendall package, using months as categorical variables. Genotype-phenotype associations for known resistance markers were evaluated with pairwise Wilcoxon texts. Potential novel markers were assessed for all SNPs with data for >50% of isolates available and at least one wild-type and mutant sample identified. The Kruskal-Wallis test with Benjamini-Hochberg correction for multiple comparisons was used to identify loci with different median drug susceptibilities between wild-type, mixed, and mutant isolates, followed by pairwise-Wilcoxon tests with Benjamini-Hochberg correction for multiple comparisons. For loci with at least 20 samples per category, a significant difference between $IC_{50}$s for wild type and mutants was required. Statistical tests were two-tailed, and significance was considered $p ≤ 0.05$.

### Reporting summary

Further information on research design is available in the Nature Portfolio Reporting Summary linked to this article.

### Data availability

Raw sequencing reads are available in the NCBI Sequence Read Archive (BioProject PRJNA850445). K13 sequences for samples genotyped using dideoxy sequencing are available in GeneBank using accession numbers: PV933992 - PV934159. MIP probes and PCR primers used in this study are available in Supplemental Table 9 of reference [24] at: https://github.com/PJRosenthalLab/2022_Tumwebaze_NatCom/blob/main/Supplemental_Table_9_Design%20of%20drug%20resistance%20MIP%20panel.xlsx. Source data for all figures and tables are provided as a Source Data file Source data are provided with this paper.

### Code availability

MIPWrangler and MIPTools software is available on GitHub (https://github.com/bailey-lab).

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

## Acknowledgements

The study was funded by the National Institutes of Health (R01AI075045, PJR; R01AI173557, MDC; U19AI089674, PJR; R01AI117001, PJR; R01AI139179, PJR; and D43TW010526, PJR), the Medicines for Malaria Venture (RD/15/0001; PJR), and the Gates Foundation (INV-035751; MDC). We thank study participants and staff members of the clinics where samples were collected. We thank Patrick Angutoko, Jackson Asiimwe, Evans Muhanguzi, Solomon Opio, Innocent Tibagambirwa, Frida G. Ceja, Shreeya Garg, and Sevil Chelebieva for performance of laboratory studies in Uganda; Bienvenu Nsengimaana, David Giesbrecht, Rebecca Crudale, Alfred Simkin, and Oriana Kreutzfeld for assistance with deep sequencing and genomic analyses; and Selina Bopp for the gift of the control KH001_053 clone. The funders of the study had no role in study design, data collection, data analysis, data interpretation, or writing of the report.

## Author contributions

M.O., S.O., P.K.T., T.K., Y.T., O.B., J.L., and M.D.C. performed parasitology and genomics experiments and archived data. JL provided administrative and logistical support. S.T., S.L.N., J.A.B., R.A.C., M.D.C., and P.J.R. provided oversight for all experiments. M.D.C. led the analysis of the data. All authors contributed to the writing of the manuscript. All authors had full access to all the data, and the corresponding authors had final responsibility for the decision to submit for publication.

## Competing interests

The authors declare no competing interests.
