## [Transparent Peer Review file · Nature Communications]

Changes in susceptibility of *Plasmodium falciparum* to antimalarial drugs in Uganda over time: 2019-2024

Corresponding Author: Dr Philip Rosenthal

Version 0:

Reviewer comments:

Reviewer #1

(Remarks to the Author)

Okitwi, Orena et al. present results from painstaking work done to determine the ex vivo susceptibility to multiple drugs of over a thousand isolates collected in Uganda, a hotspot for antimalarial drug resistance. The data presented are a significant and useful addition to the literature on the topic. The manuscript is clear and very well-written.

The points I would like to raise are mainly suggestions for additional analyses which may help answer some of the questions raised in the article. Some of these suggestions may be beyond the scope of this work but it would be interesting to have the authors' responses to all these points. If further analyses are unfeasible, the authors may wish to include these as discussion points in the manuscript.

1. Reduced lumefantrine susceptibility associated with pfk13 mutations: The association between delayed parasite clearance (ART-R) and certain pfk13 mutations has been observed quite consistently across malaria-endemic regions. Artemether-lumefantrine efficacy has mostly been sustained even in contexts where ART-R parasites are prevalent. Given these previous observations, the association (not causal relation, as the authors have acknowledged) between reduced lumefantrine susceptibility and pfk13 C469Y and A675V mutations is intriguing. Is it possible that this finding is a statistical artefact given that nearly all isolates had the wild-type N86 allele? Perhaps this is why the Benjamini-Hochberg correction (I'm afraid I'm not familiar with this) was applied, i.e., to determine whether the pfk13 mutations are independent predictors of reduced lumefantrine susceptibility? If so, this needs to be made explicit in the methods and/or results. If not, it would be worthwhile to verify the association, though it may be difficult given that there are only 7 (4 mut, 3 mixed) isolates with 86Y. Without knowing which of those are also pfk13 mutants, it is difficult to draw a conclusion.

A possibly related point (since pfmdr1 N86 prevalence is 100% 2019 onwards) - on lines 110-1, it is indicated that for some analyses, the time period was extended to samples collected before 2019 - it should be made clear for which analyses this was done.

2. Reduced DHA susceptibility associated with pfmdr1 Y500N: Similar to the point above - it would be important to check whether it mostly or always co-occurs with one of the resistance-conferring pfk13 mutations. If this is the case, the association seen may simply be with the pfk13 mutation and pfmdr1 Y500N is not an independent predictor of reduced DHA susceptibility.

3. In vivo assessment of reduced lumefantrine susceptibility: The sites where the samples for this study were collected are also where numerous clinical trials were conducted, especially in eastern Uganda. Was there any overlap between patients included in those trials and those included in the study presented here? If so, would it be possible to include treatment failure (in vivo phenotype) with the in vitro assay results to strengthen the analyses?

Other minor points:

Line 193: Please clarify how the 'Z factor' was calculated.

Lines 161-2: The cut-off for calling pfmdr1 or pfpm2/3 amplification is indicated as 1.7 (which seems high) whereas in Supp. Table 4 it is shown as 1.5 (which is more commonly applied). Please confirm/correct the actual cut-off used. If 1.7 was indeed used, would there be any amplifications detected with the lower cut-off of 1.5?

Supp. Table 5: There appears to be a typo in the prevalence over time for pfApiAP1 - the prevalence is identical over the two periods but the p-value is 2.9×10^{-22} .

Reviewer #2

(Remarks to the Author)

This is an interesting study that conducted ex vivo surveillance of nine drugs in a large number of Plasmodium falciparum isolates (>1000) from Uganda, alongside deep sequencing to identify resistance markers in candidate genes. Overall, the results indicate improved susceptibility to chloroquine but declining susceptibility to lumefantrine, mefloquine, and DHA, with resistance-associated genotypes following the same trends.

My main concerns are:

1. The clinical consequences of these changes remain uncertain. There is little description of clinical data.
2. The number of samples with mutant alleles is small, making it difficult to establish statistically significant associations for some mutations (Figure 4).
3. There is considerable phenotypic variation (Figure 4, Associations between genotypes of interest and ex vivo drug susceptibility), yet the text does not describe the standard deviation, which would help contextualize this variation.
4. The study lacks whole-genome data, particularly for samples with phenotypic data. Including such data could have facilitated the identification of novel molecular markers in candidate genes, especially in cases where ACT resistance is not associated with K13.
5. The only publicly available genotypic data is for K13. Shouldn't the full dataset be open access? The requirement to request permission to access the data raises concerns about transparency and reproducibility.

More:

Abstract:

The abstract needs improvement. Specifically, it does not mention the number of candidate genes screened and does not highlight any significant associations. Also, should clearly indicate that the clinical consequences of these changes remain uncertain

Introduction:

The statement needs clearer context and explanation "but a recent trial at a site without known ART-R at the time of the study reported <90% genotype-corrected efficacy for artemether-lumefantrine,²⁷ although genotyping to assign outcomes is challenging in such high transmission regions.

Methods:

Isolates for study

It is unclear how many new samples were collected specifically for this study. Please clarify this in the first paragraph.

The text states: "1668 samples collected since 2016; results for samples collected from June 2016 to July 2019 were published previously." This wording is somewhat confusing. How many of these samples are unique to this study? Clearly specifying this will improve clarity.

Results:

- 1-Clarify the number of new samples
- 2- Provide an explanation of the mean Z factor in the Methods section, including the threshold values that define a robust assay.
- 3- Clarify quinine potency statement. The text states: "higher for quinine (115 nM), which is typically less potent than the other studied compounds." Explain why quinine is typically less potent. Does this refer to its mechanism of action or historical data?
- 4-Report the mean and standard deviation whenever IC_{50} values are mentioned to provide statistical context.
- 5- Address the dramatic changes in IC_{50} values. The observed reductions for chloroquine and MDAQ (IC_{50} from 248 to 12.6 nM for chloroquine and 76.9 to 7.8 nM for MDAQ) are striking. Were these values normalized against controls? What are the standard deviations? Have similar trends been observed in other countries?
- 6- When discussing mutation prevalence, include exact frequencies to improve readability. For example: "Prevalences of the PfCRT K76T and PfMDR1 N86Y mutations, which are associated with resistance to chloroquine and amodiaquine, have been decreasing and were very low in recent years in our studied isolates." → Add numerical values. Similarly, for: "Prevalences of two other common PfMDR1 mutations were similar to those reported previously."
- 7- Report prevalence separately for individual and combined mutations (same haplotype) in the sentence "These mutations were at moderate prevalence in northern Uganda at the time of our first collections in 2021 (combined prevalence 30.5%)."
- 8- Report nonsynonymous mutations in candidate genes. Since multiple candidate genes were analyzed, it is important to report any nonsynonymous mutations found in these genes. Analyze whether allele frequencies have changed over time.

Discussion:

- 1-Clarify the meaning of "modest" in the context of changes in DHA and lumefantrine susceptibilities. Does it refer to statistical significance?
- 2-Emphasize the limited number of samples with mutant alleles more strongly, as this impacts the robustness of

associations.

3- Highlight the large standard deviations observed for wild-type samples, as this may suggest variability in the assay or biological differences that need further investigation.

Data sharing :

I am not happy with the fact that only K13 data is available. All data should be available , at least raw data.

Tables: Instead of reporting prevalence, please present frequency values. Include the total sample size (n) analyzed per year in each row. Without the total n, it is difficult to interpret prevalence values. For example, "111" is unclear without knowing the denominator.

Reviewer #3

(Remarks to the Author)

Okitwi et al report on their longitudinal assessment of *P. falciparum* ex vivo drug susceptibilities and genotypes in Eastern and Northern Uganda using isolates collected in 2019-24 and 2021-24 (respectively), also merging in previously published results representing eastern Uganda isolates from 2016-19.

Ex vivo susceptibilities were measured via 72 h growth inhibition assay (+ 66 h RSA for a subset of isolates) and genotyping was performed via 155-target molecular inversion probe panel targeting 80 genes of interest. Results indicate geographic susceptibility and genotype differences, and generally suggest that ex vivo susceptibilities to DHA and the ACT partner drug lumefantrine have decreased over time. Declining ex vivo susceptibilities appear to associate with PfK13 C469Y and A675V mutation prevalence and suggest that clinical efficacy may be decreasing over time.

I think this paper is a very valuable summarization of malaria parasite phenotype/genotype changes over time, and emphasizes the importance of molecular surveillance programs.

It represents a great amount of laboratory work and I find it mostly very well written and illustrated.

Only modest revisions seem necessary in my opinion, mostly for clarity purposes (details listed below) and to address the fact that minimal attention has been placed on describing possible study limitations, especially how changes in laboratory source / merging in previously published results may affect ex vivo susceptibility and genotype comparisons over time and between locations (some details below). I think many readers will wonder whether substantial shifts in ex vivo susceptibilities around 2021 (e.g., for piperazine, pyronaridine, and mefloquine) could have any artefactual basis. Same for some sudden shifts in IC50 around x-axis tick 2022 for chloroquine, pyronaridine, MDAQ, mefloquine in N. Uganda (Supp. Fig2)? I could not say due to lack of my experience with these assays but also a lack of detail regarding methods execution at different labs at specific timepoints in the manuscript (the info is mentioned but in unclear fashion).

Also more broadly, I would have appreciated a bit more discussion on why geographic differences within Uganda are thought to occur for selected drug resistance-associated mutations (e.g. PfK13 C469Y) and phenotypes. E.g., is this an incipient, epidemic process of mutant emergence/importation and expansion, limited by landscape barriers, drug regime differences, and human mobility patterns between N/E Uganda?

Specific line comments:

32-34 (abstract): IC50 values seem unnecessarily vaguely stated: can you not specify what values you mean by low nanomolar vs. higher values? E.g.:

".....low nanomolar (1.5 – 15.2 nM) and higher for quinine (115 nM) and pyrimethamine (35,100 nM)....." ?

34 (abstract): 'improved' seems to be subjective -> maybe use objective words like increased etc. throughout the manuscript?

36-42 (abstract): again relatively ambiguously stated: "Changes in prevalences of known markers ... followed the same patterns." You have nice details in the main text which could be concisely incorporated here to make the abstract more compelling/informative.

41 (abstract): similar to line 34 comment... activities -> susceptibilities? Try to be more constant/methodical with wording.

61: "opposite effect" --- please reword this sentence for clarity.

58-80: throughout this section it is a bit unclear whether effects of mutations on susceptibility represent combinatorial mutation effects or single mutation effects.

111: I would indicate the extra 1668 published samples represent Eastern Uganda (?).

201: awkward explanation: "more marked changes in susceptibilities occurred in eastern Uganda, consistent with a longer period of sample collection in that area"

(the marked shift in E. Uganda occurs between ca. 2016 to 2020 (a period mostly not sampled in the North) and ca. 2021 to 2024.

217: Of the 1,113 isolates -> typo? 1114?

241: I would specify this as MIPS based sequencing

Fig. 1: Labeling could be more detailed, e.g., include Agogo and other key district labeling, also colors changed to not match water, etc.

It could also be explicitly labeled which sample sets had susceptibilities tested in Tororo hospital vs. Kalongo hospital, as methods suggest Agogo samples were first analyzed at Tororo (2021-2022) but that sentence is quite hard to follow.

Fig. 2: Paired plots could have y axis scale matched and it might be useful to include some minor shading/horiz. line etc to indicate where e.g., 10 nM etc. is on all plot pairs, since different pairs use different scales. For the tenth plot, I would additionally indicate that RSA corresponds to DHA challenge, for less familiar readers. Also, do x-axis tick represent January of each year? It would be nice to see time (months) in better detail.

Fig. 3: Here or in methods it would be nice to know which amino acid ranges are covered by the MIPS panel? E.g. are the three PfK13 positions you have in the plots those that were polymorphic in the sample set although you do have results for the whole protein / propeller domain? Or are other PfK13 positions not shown because they do not exist/fail in the assay? Also, how does the figure's y-axis handle mutation detection within polyclonal samples?

Supp. Fig. 2. What do the abrupt upward IC50 shifts around x-axis tick 2022 mean for chloroquine, pyronaridine, MDAQ, mefloquine?

Supplementary Table 2. Unclear if this is a comparison between Tororo and Kalongo laboratory results?

A bit more labeling of supp. plots/table legends/subtitles could help throughout this manuscript.

Supplementary Table 4. Could you add columns for the control results (Dd2, KH001_053) so it is easier to understand what a value like 1.6 means?

Supplementary Table 7. Pyrimethamine spello in title

Version 1:

Reviewer comments:

Reviewer #1

(Remarks to the Author)

The comments/queries I had raised previously have been satisfactorily addressed. I only had a couple of additional minor comments for the authors' consideration:

1. Supplementary table 7 / pfmdr1 500N: since only pure mutants or pure wild-types were considered for these analyses, it would be interesting to add the results of the ex vivo RSA (less reliable in ex vivo assays with mixed infections as the authors reasonably posit) to the table. A similar trend, indicating an association between pfmdr1 500N and reduced DHA susceptibility, will presumably be observed. If not, it may be useful to add this point to the discussion.

2. Association of pfk13 C469Y, A675V with reduced lumefantrine susceptibility: The authors have explicitly acknowledged that there is no direct evidence implicating mutations in pfk13 with lumefantrine susceptibility. What they hint at but do not make as explicit (perhaps deliberately, in order not to be overly speculative) is the possibility that parasites with these mutations in pfk13 have acquired other mutation(s) which confer reduced susceptibility to lumefantrine. This would at least partially explain why these two mutations are increasingly prevalent, similar to the KEL1/PLA1 lineage which was predominant in the Greater Mekong Subregion. The authors may want to emphasise this point in the discussion section.

Reviewer #2

(Remarks to the Author)

I am happy with the responses to the reviewers.
I agree with the paper being accepted.

Reviewer #3

(Remarks to the Author)

This an excellent work profiling ex vivo susceptibility to multiple drugs in Uganda at large scale. The authors have clarified the methods details and incorporated further commentary on the (uncertain) clinical consequences of the observed changes in artemether-lumefantrine susceptibility in the region.

I am fully satisfied with the thorough responses.

Responses to review comments:

Notes from the authors:

- 1) Line numbers in our responses refer to the tracked version of our MS.
- 2) We moved Methods to after the Discussion; to avoid a mess we did not track this change.
- 3) Also to avoid messiness we did not track changes to tables.

REVIEWER COMMENTS

Reviewer #1 (Remarks to the Author):

Okitwi, Orena et al. present results from painstaking work done to determine the ex vivo susceptibility to multiple drugs of over a thousand isolates collected in Uganda, a hotspot for antimalarial drug resistance. The data presented are a significant and useful addition to the literature on the topic. The manuscript is clear and very well-written.

The points I would like to raise are mainly suggestions for additional analyses which may help answer some of the questions raised in the article. Some of these suggestions may be beyond the scope of this work but it would be interesting to have the authors' responses to all these points. If further analyses are unfeasible, the authors may wish to include these as discussion points in the manuscript.

1. Reduced lumefantrine susceptibility associated with pfk13 mutations: The association between delayed parasite clearance (ART-R) and certain pfk13 mutations has been observed quite consistently across malaria-endemic regions. Artemether-lumefantrine efficacy has mostly been sustained even in contexts where ART-R parasites are prevalent. Given these previous observations, the association (not causal relation, as the authors have acknowledged) between reduced lumefantrine susceptibility and pfk13 C469Y and A675V mutations is intriguing. Is it possible that this finding is a statistical artefact given that nearly all isolates had the wild-type N86 allele? Perhaps this is why the Benjamini-Hochberg correction (I'm afraid I'm not familiar with this) was applied, i.e., to determine whether the pfk13 mutations are independent predictors of reduced lumefantrine susceptibility? If so, this needs to be made explicit in the methods and/or results. If not, it would be worthwhile to verify the association, though it may be difficult given that there are only 7 (4 mut, 3 mixed) isolates with 86Y. Without knowing which of those are also pfk13 mutants, it is difficult to draw a conclusion.

RESPONSE: The N86 allele WT is associated with slightly decreased LM activity, as well documented in the literature (including our older reports), but changes in recent years, in the setting of near 100% prevalence of the WT allele, indicate that other genetic changes are driving decreasing LM susceptibility. The Benjamini-Hochberg correction was applied to address concerns about multiple comparisons leading to chance identification of significant associations. In response to a specific query, we could not test associations between PfMDR1/PfK76 mutations and PfK13 mutations, as these never coincided. This was explained as follows: "The few PfMDR1 N86Y and PfCRT K76T mutations identified were seen exclusively with PfK13 wild-type sequences." (line 184)

A possibly related point (since pfmdr1 N86 prevalence is 100% 2019 onwards) - on lines 110-1, it is indicated that for some analyses, the time period was extended to samples collected before 2019 - it should be made clear for which analyses this was done.

RESPONSE: We apologize for not being clearer. We extended analyses back to 2016 only for our search for associations between genotypes and drug susceptibility phenotypes (the data described in Figure 4). This was explained when we presented these data in Results (line

168-171), but to improve clarity we have more clearly explained this point in Methods (line 306-309): “To increase sample size for genotype-phenotype association studies we extended our analyses to include 1668 samples collected since 2016, including results for 554 samples which were published previously, all from eastern Uganda and collected from June, 2016 to July, 2019.”

2. Reduced DHA susceptibility associated with *pfmdr1* Y500N: Similar to the point above - it would be important to check whether it mostly or always co-occurs with one of the resistance-conferring *pfk13* mutations. If this is the case, the association seen may simply be with the *pfk13* mutation and *pfmdr1* Y500N is not an independent predictor of reduced DHA susceptibility.

RESPONSE: This is a good point. We now include data on associations between relevant PfK13 mutations and the PfMDR1 500N and PfCARL D611N mutations, which were associated with decreased lumefantrine and DHA susceptibility, respectively. Indeed, we added a paragraph to the Results section and a new Supplementary Table 7 to fully address this matter. The new paragraph is as follows (line 184-194):

“To consider the interplay between PfK13 mutations and candidate resistance markers, we assessed shared prevalence. The few PfMDR1 N86Y and PfCRT K76T mutations identified were seen exclusively with PfK13 wild-type sequences. Considering only pure mutant and pure wild-type genotypes to avoid haplotype assumptions, PfCARL D611N was seen with PfK13 wild-type, C469Y mutant, and A675V mutant sequences, while PfMDR1 Y500N was seen with PfK13 wild-type and C469Y mutant sequences. Susceptibilities to lumefantrine were lowest in the presence of PfK13 C469Y and/or A675V mutations with either wild-type or mutant alleles at PfCARL D611N and for DHA were lowest in the presence of the PfK13 C469Y mutation with either wild-type or mutant alleles at PfMDR1 Y500N, although few samples were available for some comparisons (Supplementary Table 7). The mutations were also associated with decreased susceptibilities in the absence of PfK13 mutations.”

3. In vivo assessment of reduced lumefantrine susceptibility: The sites where the samples for this study were collected are also where numerous clinical trials were conducted, especially in eastern Uganda. Was there any overlap between patients included in those trials and those included in the study presented here? If so, would it be possible to include treatment failure (in vivo phenotype) with the in vitro assay results to strengthen the analyses?

RESPONSE: There was no overlap. Samples for this study were all collected from patients not enrolled in clinical trials who presented to participating health centers with uncomplicated malaria.

Other minor points:

Line 193: Please clarify how the 'Z factor' was calculated.

RESPONSE: This information was referenced, but it is now added to our Methods section, as follows. “To assess well-to-well variability and signal-to-noise ratios in the fluorescence readout of ex-vivo assays, Z factors were calculated as: $Z=1 - ([3 \times SD_{\text{infected drug free}} + 3 \times SD_{\text{uninfected}}] / [\text{mean}_{\text{infected drug free}} - \text{mean}_{\text{uninfected}}])$.” (line 369-371)

Lines 161-2: The cut-off for calling *pfmdr1* or *pfpm2/3* amplification is indicated as 1.7 (which seems high) whereas in Supp. Table 4 it is shown as 1.5 (which is more commonly applied). Please confirm/correct the actual cut-off used. If 1.7 was indeed used, would there be any amplifications detected with the lower cut-off of 1.5?

RESPONSE: As noted, we included data with a cut-off of 1.5 in the table, but used a slightly higher cut-off when discussing the data in the text. To address this concern we changed our description in Results to the following. “Definitive increased copy number of *pfmdr1*³² or

plasmepsin 2/3,^{9, 33} which has been associated with decreased susceptibility to lumefantrine and mefloquine or piperazine, respectively, was not observed, with the vast majority of copy numbers measured at ≤ 1.5 , and copy number of 1.6 seen for 1/661 isolates for *pfmdr1* and 4/445 isolates for *plasmepsin 2* (Supplementary Table 4).” (line 149-153)

Supp. Table 5: There appears to be a typo in the prevalence over time for pfApiAP1 - the prevalence is identical over the two periods but the p-value is 2.9×10^{-22} .

RESPONSE: Thanks for noting this error. We have corrected the table and proofread all tables to assure accuracy.

Reviewer #2 (Remarks to the Author):

This is an interesting study that conducted ex vivo surveillance of nine drugs in a large number of *Plasmodium falciparum* isolates (>1000) from Uganda, alongside deep sequencing to identify resistance markers in candidate genes. Overall, the results indicate improved susceptibility to chloroquine but declining susceptibility to lumefantrine, mefloquine, and DHA, with resistance-associated genotypes following the same trends.

My main concerns are:

1. The clinical consequences of these changes remain uncertain. There is little description of clinical data.

RESPONSE: This was not a clinical study, but we are very interested in the clinical consequences of our findings. Indeed, correlations between molecular genotypes, lab phenotypes, and clinical findings have been complex, and we think often misunderstood. We expanded a brief consideration of clinical consequences in the Introduction (see response to Rev #2, Introduction below). We also reorganized the Discussion, moving a few sentences and adding a new paragraph that offers an extensive discussion regarding our understanding of the clinical consequences of changes in ex vivo susceptibilities of malaria parasites to key antimalarial drugs (lines 241-257): “The clinical consequences of the observed changes in *P. falciparum* susceptibility to artemether-lumefantrine in Uganda remain uncertain. Most recent trials have demonstrated excellent treatment efficacies for artemether-lumefantrine.^{5, 11, 27} However, studies in Angola,³⁶ Burkina Faso,³⁷ Democratic Republic of Congo,³⁸ Tanzania,³⁹ and Uganda²⁸ have shown genotype-corrected treatment efficacies for AL <90% at certain sites. In all cases the prevalence of ART-R-mediating PfK13 mutations was very low at the times of the studies, so the sub-optimal treatment efficacy was not due in a straightforward manner to ART-R. Further, these results must be interpreted with caution, as assignment of treatment outcomes using varied molecular genotyping methods is inexact, especially in areas of very high malaria transmission.^{40, 41, 42} In addition, *P. falciparum* infections recently acquired in Africa and evaluated elsewhere have demonstrated decreased in vitro activity of lumefantrine, enhanced RSA survival, and multiple failures of therapy with artemether-lumefantrine both with^{43,44} and without^{44, 45} known ART-R-mediating PfK13 mutations. Taken together, although our understanding of this area remains incomplete, the combination of evidence for decreasing activities of both components of artemether-lumefantrine in Uganda in prior studies^{23, 24} and this report and of sub-optimal clinical efficacy for artemether-lumefantrine in multiple countries is concerning.”

2. The number of samples with mutant alleles is small, making it difficult to establish statistically significant associations for some mutations (Figure 4).

RESPONSE: We agree, and we addressed this limitation in the sentence describing results for mutations that are now uncommon: “As described previously,⁸ the PfMDR1 N86Y wild-type allele was associated with decreased susceptibility to lumefantrine and mefloquine

and the PfCRT K76T wild-type allele with decreased susceptibility to lumefantrine, although analyses were limited by low prevalence of mutant genotypes.” (line 174-178).

3. There is considerable phenotypic variation (Figure 4, Associations between genotypes of interest and ex vivo drug susceptibility), yet the text does not describe the standard deviation, which would help contextualize this variation.

RESPONSE: Standard deviation is not the appropriate summary statistic for our data, as we utilized medians, which we believe are more representative of the data than means, which can be confounded by outlier values. We included error bars in Figure 4, and we have added an explanation of what they represent (which is the default representation of these data in R). Specifically, we added the following to the Figure 4 legend: “Center bounds of boxes correspond to the medians, and minimal and maximal bounds correspond to 25th and 75th percentiles, respectively. Whiskers extend to extreme values no further than 1.5x the IQR from the 25th or 75th percentiles.”

4. The study lacks whole-genome data, particularly for samples with phenotypic data. Including such data could have facilitated the identification of novel molecular markers in candidate genes, especially in cases where ACT resistance is not associated with K13.

RESPONSE: We are pursuing whole genome sequencing of selected isolates, but this work is beyond the scope of this MS. We have added mention of this ongoing activity: “Ongoing whole genome sequencing of select isolates may identify additional potential mediators of altered drug susceptibility.” (line 275)

5. The only publicly available genotypic data is for K13. Shouldn't the full dataset be open access? The requirement to request permission to access the data raises concerns about transparency and reproducibility.

RESPONSE: This appears to be a misunderstanding. All sequence data are available as indicated in the Methods section: “Sequencing reads are available in the National Center for Biotechnology Information Archive (BioProject PRJNA850445)” (line 357).

More:

Abstract:

The abstract needs improvement. Specifically, it does not mention the number of candidate genes screened and does not highlight any significant associations. Also, should clearly indicate that the clinical consequences of these changes remain uncertain

RESPONSE: The reviewer may not be aware that the limit for the abstract is 150 words, limiting our ability to present detailed data. To address the specific concerns we have added two clauses; we hope that the editors will allow these few extra words. First, in describing the study, we add mention of the number of genes sequenced: “We characterized ex vivo susceptibilities to nine drugs of isolates collected from individuals presenting with uncomplicated falciparum malaria in eastern (2019-2024) and northern (2021-2024) Uganda and performed deep sequencing, with analysis of 80 Plasmodium falciparum genes...”. (line 30) Second, considering clinical consequences, we added a clause to the last sentence of the abstract: “Decreasing activities of lumefantrine and DHA suggest potential loss of efficacies of leading regimens, although the clinical consequences of these changes are, to date, uncertain.” (line 43)

Introduction:

The statement needs clearer context and explanation ” “but a recent trial at a site without known ART-R at the time of the study reported <90% genotype-corrected efficacy for artemether-

lumefantrine,²⁷ although genotyping to assign outcomes is challenging in such high transmission regions.

RESPONSE: To provide more context we have expanded this sentence to summarize a set of trials with varied responses to therapy, citing a short review that covered this topic in detail in addition to citing the key recent paper describing a trial in Uganda, as follows. “Most clinical trials have shown excellent efficacies of leading ACTs in Uganda,^{25, 26} but there have been exceptions (at sites without known ART-R at the times of the studies) with artemether-lumefantrine genotype-corrected treatment efficacies <90%,²⁷ including a recent trial in Uganda,²⁸ although genotyping to assign outcomes is challenging in such high transmission regions.” (line 89-95) We also added a paragraph in the Discussion addressing the clinical relevance of our findings in detail (see response to point #1 from this reviewer above).

Methods:

Isolates for study

It is unclear how many new samples were collected specifically for this study. Please clarify this in the first paragraph.

The text states: “1668 samples collected since 2016; results for samples collected from June 2016 to July 2019 were published previously.” This wording is somewhat confusing. How many of these samples are unique to this study? Clearly specifying this will improve clarity.

RESPONSE: We apologize for not being clearer. We extended analyses back to 2016 only for our search for associations between genotypes and drug susceptibility phenotypes (the data described in Figure 4). This was explained when we presented these data in Results (line 168-171), but to improve clarity we have more clearly explained this point in Methods (line 306-309): “To increase sample size for genotype-phenotype association studies we extended our analyses to include 1668 samples collected since 2016, including results for 554 samples which were published previously, all from eastern Uganda and collected from June, 2016 to July, 2019.”

Results:

1-Clarify the number of new samples

RESPONSE: Sample collection is clearly described in Methods, but to add clarity we added mention of the initial date of collection also in the first sentence of Results, as follows. “Of 1,297 *P. falciparum* isolates collected since July, 2019...” (line 105).

2- Provide an explanation of the mean Z factor in the Methods section, including the threshold values that define a robust assay.

RESPONSE: This information was referenced, but it is now added to our Methods section, as follows. “To assess well-to-well variability and signal-to-noise ratios in the fluorescence readout of ex-vivo assays, Z factors were calculated as: $Z=1 - ([3 \times SD_{\text{infected drug free}} + 3 \times SD_{\text{uninfected}}] / [\text{mean}_{\text{infected drug free}} - \text{mean}_{\text{uninfected}}])$.” (line 369-371)

3- Clarify quinine potency statement. The text states: “higher for quinine (115 nM), which is typically less potent than the other studied compounds.” Explain why quinine is typically less potent. Does this refer to its mechanism of action or historical data?

RESPONSE: The point of our comment (“which is typically less potent than the other studied compounds”) was simply to note that our results for quinine were not surprising. Different compounds have different potencies, e.g. a typical IC₅₀ for pyronaridine is 1-2 nM, for chloroquine is ~10-20 nM (for a sensitive parasite), and for quinine is ~50-100 nM or higher. The biology of the antimalarial activity of quinine is poorly understood, so we cannot explain why quinine is less potent. But, differences in potencies between different antimicrobial drugs, including different antimalarials, is common knowledge.

4-Report the mean and standard deviation whenever IC₅₀ values are mentioned to provide statistical context.

RESPONSE: Standard deviation is not the appropriate summary statistic for our data, as we utilized medians, which we believe are more representative of the data than means, which can be confounded by outlier values. The data presented included statistical analyses, as described in Methods and figure/table legends.

5- Address the dramatic changes in IC₅₀ values. The observed reductions for chloroquine and MDAQ (IC₅₀ from 248 to 12.6 nM for chloroquine and 76.9 to 7.8 nM for MDAQ) are striking. Were these values normalized against controls? What are the standard deviations? Have similar trends been observed in other countries?

RESPONSE: Controls were run regularly, as explained in the text (line 346: “Control *P. falciparum* Dd2 (MRA-156) and 3D7 (MRA-102) strains from MR4/BEI Resources were maintained in culture, synchronised with a magnetic column, and assayed, beginning at the ring-stage, approximately monthly.”), and results for controls were consistent, such that normalization was not necessary (and likely would have added confusion). The results are not surprising, as malaria parasites have lost markers of resistance to CQ and AQ over the last two decades, and they are consistent with other reports from our group and others showing dramatic improvements in activities for CQ and AQ with loss of known resistance mediators (e.g. see reference 23: Tumwebaze, et al., Lancet Microbe, 2021, Fig. 4).

6- When discussing mutation prevalence, include exact frequencies to improve readability. For example:

“Prevalences of the PfCRT K76T and PfMDR1 N86Y mutations, which are associated with resistance to chloroquine and amodiaquine, have been decreasing and were very low in recent years in our studied isolates.” → Add numerical values.

Similarly, for: “Prevalences of two other common PfMDR1 mutations were similar to those reported previously.”

RESPONSE: We appreciate this request, but in writing this report we aimed for clear presentation, with relatively few numerical values in the text, and all necessary quantitative details in tables. Adding numerical values to all descriptions in the text will add a great deal of clutter, and we do not think that it will improve communication.

7- Report prevalence separately for individual and combined mutations (same haplotype) in the sentence “These mutations were at moderate prevalence in northern Uganda at the time of our first collections in 2021 (combined prevalence 30.5%).”

RESPONSE: We made this change (line 161), and in doing so corrected an error (we accidentally reported prevalence for only one of the mutations): “These mutations were at moderate prevalence in northern Uganda at the time of our first collections in 2021 (prevalence 30.5% for C469Y and 8.5% for A675V), with stable prevalence since that time.” Thanks to the reviewer for noticing this.

8- Report nonsynonymous mutations in candidate genes. Since multiple candidate genes were analyzed, it is important to report any nonsynonymous mutations found in these genes. Analyze whether allele frequencies have changed over time.

RESPONSE: It seems unwieldy and not terribly valuable to report all mutations seen in 80 genes from >1000 samples. For anyone interested, this information can be gleaned from submitted genomic information. Rather, for this MS, we focus on mutations that were associated with alterations in drug susceptibility. Data for these, including changes in allele frequencies over time, are in Supplementary tables 5, 6, and 8. Criteria for inclusion of mutations in these

tables is included in Methods; we added a statement in this regard to the Supplementary Table 5 legend as follows: “The Kruskal-Wallis test with Benjamini-Hochberg correction for multiple comparisons was used to identify loci with different median drug susceptibilities between WT, mixed, and mutant isolates, followed by pairwise-Wilcoxon tests with Benjamini-Hochberg correction for multiple comparisons. Loci with at least 20 samples per category and significant differences between IC₅₀s for WT and mutants are shown.”

Discussion:

1-Clarify the meaning of "modest" in the context of changes in DHA and lumefantrine susceptibilities. Does it refer to statistical significance?

RESPONSE: We believe that it is appropriate in a discussion to use adjectives, not only numbers, to describe complex phenomena. Adding numbers to this brief summary of data in the first paragraph of the Discussion will obscure a simple presentation. The specific meaning of “modest” will be apparent to any reader who looks at the actual data provided in tables.

2-Emphasize the limited number of samples with mutant alleles more strongly, as this impacts the robustness of associations.

RESPONSE: It is unclear which alleles the reviewer is talking about here. Some mutations were common, and some uncommon. The number of samples studied is described in tables. We addressed limited numbers of mutations available to assess some associations, e.g.: “As described previously,⁸ the PfMDR1 N86Y wild-type allele was associated with decreased susceptibility to lumefantrine and mefloquine and the PfCRT K76T wild-type allele with decreased susceptibility to lumefantrine, although analyses were limited by low prevalence of mutant genotypes.” (line 174-178)

3- Highlight the large standard deviations observed for wild-type samples, as this may suggest variability in the assay or biological differences that need further investigation.

RESPONSE: As noted above, we have not reported SDs. We agree that values for WT samples vary, evidence that identified genetic markers do not explain all variability seen in our *P. falciparum* isolates. We are continuing to study this area. In the MS we added the following sentences to the Discussion to address this matter: “The variability of drug susceptibilities for isolates without known drug resistance-mediating mutations suggests that additional genetic polymorphisms impact on susceptibility. Ongoing whole genome sequencing of select isolates may identify additional potential mediators of altered drug susceptibility. (line 273-276)

Data sharing :

I am not happy with the fact that only K13 data is available. All data should be available , at least raw data.

RESPONSE: This appears to be a misunderstanding. All sequence data are available as indicated in the Methods section: “Sequencing reads are available in the National Center for Biotechnology Information Archive (BioProject PRJNA850445)” (line 357).

Tables: Instead of reporting prevalence, please present frequency values. Include the total sample size (n) analyzed per year in each row. Without the total n, it is difficult to interpret prevalence values. For example, "111" is unclear without knowing the denominator.

RESPONSE: Presumably the reviewer is referring to Supplementary Table 3. The request for “n” is unclear, as N, the denominator (the total number of samples with a valid genotype), is provided for each year on the table. All isolates were genotyped. Due to limitations of our sample set, we cannot reliably provide frequency (the mean percentage of a polymorphism within the full set of samples), rather than prevalence (the percentage of samples that contained a polymorphism), but it is not clear how this different metric will add notably to the

value of our prevalence data.

Reviewer #3 (Remarks to the Author):

Okitwi et al report on their longitudinal assessment of *P. falciparum* ex vivo drug susceptibilities and genotypes in Eastern and Northern Uganda using isolates collected in 2019-24 and 2021-24 (respectively), also merging in previously published results representing eastern Uganda isolates from 2016-19.

Ex vivo susceptibilities were measured via 72 h growth inhibition assay (+ 66 h RSA for a subset of isolates) and genotyping was performed via 155-target molecular inversion probe panel targeting 80 genes of interest. Results indicate geographic susceptibility and genotype differences, and generally suggest that ex vivo susceptibilities to DHA and the ACT partner drug lumefantrine have decreased over time. Declining ex vivo susceptibilities appear to associate with Pfk13 C469Y and A675V mutation prevalence and suggest that clinical efficacy may be decreasing over time.

I think this paper is a very valuable summarization of malaria parasite phenotype/genotype changes over time, and emphasizes the importance of molecular surveillance programs. It represents a great amount of laboratory work and I find it mostly very well written and illustrated.

Only modest revisions seem necessary in my opinion, mostly for clarity purposes (details listed below) and to address the fact that minimal attention has been placed on describing possible study limitations, especially how changes in laboratory source / merging in preciously published results may affect ex vivo susceptibility and genotype comparisons over time and between locations (some details below). I think many readers will wonder whether substantial shifts in ex vivo susceptibilities around 2021 (e.g., for piperazine, pyronaridine, and mefloquine) could have any artefactual basis. Same for some sudden shifts in IC50 around x-axis tick 2022 for chloroquine, pyronaridine, MDAQ, mefloquine in N. Uganda (Supp. Fig2)? I could not say due to lack my lack of experience with these assays but also a lack of detail regarding methods execution at different labs at specific timepoints in the manuscript (the info is mentioned but in unclear fasion).

Also more broadly, I would have appreciated a bit more discussion on why geographic differences within Uganda are thought to occur for selected drug resistance- associated mutations (e.g. Pfk13 C469Y) and phenotypes. E.g., is this an incipient, epidemic process of mutant emergence/importation and expansion, limited by landscape barriers, drug regime differences, and human mobility patterns between N/E Uganda?

Specific line comments:

32-34 (abstract): IC50 values seems unnecessarily vaguely stated: can you not specify what values you mean by low nanomolar vs. higher values? E.g.:

".....low nanomolar (1.5 – 15.2 nM) and higher for quinine (115 nM) and pyrimethamine (35,100 nM)....." ?

RESPONSE: The reviewer notes concern about potential artefactual shifts in susceptibility measures (which we share) and then requests precise reporting of IC50 values in the abstract. We chose not to include numerical summaries in the abstract for two reasons. First, the 150 word limit is daunting; including values for the 9 study drugs would have forced us to greatly shorten our already constrained summary. Second, due to imprecision and also variability in IC50 measurements between labs due for technical reasons (most striking for

lumefantrine, where assay in serum vs Albumax media can lead to 10-fold differences), it seems more appropriate to provide only general trends, with, of course, all precise values provided in detailed tables accompanying the full text.

34 (abstract): 'improved' seems to subjective -> maybe use objective words like increased etc. throughout the manuscript?

RESPONSE: We crafted our language carefully. We chose “improved” rather than “increased” because, when considering IC50 data it is easy for readers to become confused when an increased IC50 signals decreased susceptibility.

36-42 (abstract): again relatively ambiguously stated: "Changes in prevalences of known markers ... followed the same patterns." You have nice details in the main text which could be concisely incorporated here to make the abstract more compelling/informative.

RESPONSE: Again, our tight language was due to the 150 word limit for the abstract. We would love for Nat Commun to allow longer abstracts. We spent many hours crafting the abstract, and it is not clear to us how we can “concisely incorporate” added information without removing other key wording. We have already added words to the abstract at the request of another reviewer, but this request would require substantially more. If the editors will allow us extra words, we will be most happy to describe changes in prevalences of drug resistance markers in more detail in the abstract.

41 (abstract): similar to line 34 comment... activities -> susceptibilities? Try to be more constant/methodical with wording.

RESPONSE: We respectfully disagree, as the reviewer appears to have missed a subtle, but important distinction. In our abstract “susceptibility” refers to the results of our ex vivo assays and “activity” refers to the more general concept of the ability of the drug to have its desired effect. We use the terms consistently through the MS.

61: "opposite effect" --- please reword this sentence for clarity.

RESPONSE: The “opposite” (also referred to by some as “reciprocal”) effects of aminoquinolines (CQ and AQ) vs. lumefantrine/mefloquine on drug susceptibility is very well described and known in the malaria research community. The meaning of “opposite” seems clear to us, as it is explained in the sentence in question: “First, resistance to chloroquine and amodiaquine is mediated principally by the PfCRT K76T and PfMDR1 N86Y mutations, which alter drug transport..... An opposite effect is seen with lumefantrine and mefloquine, with the PfCRT K76T and PfMDR1 N86Y wild-type alleles and *pfmdr1* amplification associated with decreased susceptibility.” Thus, we respectfully request that our wording be retained.

58-80: throughout this section it is a bit unclear whether effects of mutations on susceptibility represent combinatorial mutation effects or single mutation effects.

RESPONSE: This is a good point, but it is difficult to address. Our understanding of the roles of haplotypes, rather than individual mutations, in antimalarial drug resistance is limited, and we disagree with some general summaries in the literature regarding haplotypes (e.g. we do not agree, based on extensive data, that a much-discussed 3 allele PfDHFR haplotype is a resistance mediator, but rather see that the N86Y mutation is the driver of this phenotype, with data from our lab suggesting that Y184F impacts on parasite fitness and D1246Y plays a minor role). For ART-R, to our knowledge PfK13 generally contains only one propeller domain mutation in any strain, probably because additional mutations will constrain fitness. Considering these complexities, we have only considered individual mutations. To make this clearer for readers considering PfK13 mutations, which are most relevant, we added a parenthetical clause, as follows. “ART-R manifests as delayed parasite clearance after therapy with

artemisinins and enhanced parasite survival after in vitro exposure to DHA, and is mediated principally by any of ~20 mutations (with generally only one of these mutations in an individual strain) in the *P. falciparum* kelch (PfK13) protein propeller domain.” (line 68-71)

111: I would indicate the extra 1668 published samples represent Eastern Uganda (?).

RESPONSE: Thanks for this helpful suggestion. We added clarification of the origins of these samples, as follows. “To increase sample size for genotype-phenotype association studies we extended our analyses to include 1668 samples collected since 2016, including results for 554 samples which were published previously, all from eastern Uganda and collected from June, 2016 to July, 2019.” (line 306-309)

201: awkward explanation: “more marked changes in susceptibilities occurred in eastern Uganda, consistent with a longer period of sample collection in that area” (the marked shift in E. Uganda occurs between ca. 2016 to 2020 (a period mostly not sampled in the North) and ca. 2021 to 2024.

RESPONSE: We changed the wording of this sentence to add clarity: “The more marked changes in susceptibilities occurred in eastern Uganda, likely with decreases in susceptibilities to key drugs in northern Uganda before initiation of studies in that region, as suggested by our earlier comparison of the sites (Supplementary Figures 1 and 2).²⁴” (line 121-125)

217: Of the 1,113 isolates -> typo? 1114?

RESPONSE: We corrected this error.

241: I would specify this as MIPS based sequencing

RESPONSE: Sequencing was mostly with the MIPS platform, but we resolved some ambiguities with dideoxy sequencing, as clearly described in Methods. Considering this complexity, it seems most appropriate not to rehash the somewhat complex sequencing methods in the Results section.

Fig. 1: Labeling could be more detailed, e.g., include Agogo and other key district labeling, also colors changed to not match water, etc.

It could also be explicitly labeled which sample sets had susceptibilities tested in Tororo hospital vs. Kalongo hospital, as methods suggest Agogo samples were first analyzed at Tororo (2021-2022) but that sentence is quite hard to follow.

RESPONSE: We chose not to label districts due to the complexity of our studies in E. Uganda, which included collection from three nearby sites in three different districts (Busia, Tororo, and Mbale), and even potential confusion about the fact that Busiu Health Centre is in Mbale District, though it is near Busia District (note the slightly different spelling). To address this request we changed the colors on Figure 1 to distinguish labels and lakes. We agree that this is an improvement.

Fig. 2: Paired plots could have y axis scale matched and it might be useful to include some minor shading/horiz. line etc to indicate where e.g., 10 nM etc. is on all plot pairs, since different pairs use different scales. For the tenth plot, I would additionally indicate that RSA corresponds to DHA challenge, for less familiar readers.

RESPONSE: What is most important in this figure is not differences between drugs, but differences for each drug over time. Therefore, we respectfully do not think it makes sense to use the same y-axis scale for each plot, as this will obscure data for the more potent drugs. We also do not think that these already busy figures will benefit from shading. The RSA is only conducted for DHA both in the literature and as clearly described in this MS.

Also, do x-axis tick represent January of each year? It would be nice to see time (months) in better detail.

RESPONSE: The precise date of the x-axis tick is now described in the figure legend: "X-axis tick marks indicate June 1 of each year."

Fig. 3: Here or in methods it would be nice to know which amino acid ranges are covered by the MIPS panel? E.g. are the three PfK13 positions you have in the plots those that were polymorphic in the sample set although you do have results for the whole protein / propeller domain? Or are other PfK13 positions not shown because they do not exist/fail in the assay? Also, how does the figure's y-axis handle mutation detection within polyclonal samples?

RESPONSE: Genomic coordinates for all MIP probes are included in Supplementary Table 9 of reference 24: Tumwebaze, et al, Nat Commun, 2023 (PMID 36289202), which is referenced at the appropriate place in the Methods section (line 355). The prevalence shown in this figure is the percentage of isolates with a wild-type, mixed, or mutant sequence at the indicated allele, as indicated by the color code that is defined in the legend.

Supp. Fig. 2. What do the abrupt upward IC50 shifts around x-axis tick 2022 mean for chloroquine, pyronaridine, MDAQ, mefloquine?

RESPONSE: Our reading of the data shows some year-to-year changes, but not "abrupt upward IC50 shifts." As the reviewer probably suspects, we do not have a clear explanation for the minor variations seen over time. It is noteworthy that the oldest data for the N. Uganda site were collected in an episodic manner in 2021 and 2022, before we had a laboratory in N. Uganda, and thus transported samples for only part of the year to our laboratory in E. Uganda; although these discontinuous data are less aesthetically pleasing than the continuous data from E. Uganda or later N. Uganda samples, we think they add importantly to the body of results.

Supplementary Table 2. Unclear if this is a comparison between Tororo and Kalongo laboratory results?

RESPONSE: As noted in the table, it compares isolates collected in N. vs. E. Uganda. As clearly described in Methods, the older assays (2021-22) for N. Uganda isolates were performed in our E. Uganda lab: "We added collection of samples from patients with uncomplicated malaria in Agago District, in northern Uganda, first with transport of samples from Patongo Health Centre IV to our Tororo laboratory from May to July, 2021 and April to July, 2022, and since January, 2023 with samples from Patongo Health Centre IV and Dr. Ambrosoli Memorial Hospital, Kalongo assessed in a new laboratory at the hospital."

A bit more labeling of supp. plots/table legends/subtitles could help throughout this manuscript.

RESPONSE: We have added to figure and table legends in response to other reviewer requests, as detailed above and shown in the tracked version of the MS.

Supplementary Table 4. Could you add columns for the control results (Dd2, KH001_053) so it is easier to understand what a value like 1.6 means?

RESPONSE: We added values for 3 control strains to Supplementary Table 4.

Supplementary Table 7. Pyrimethamine spello in title

RESPONSE: Thanks. We corrected this typo.

Responses to review comments (second revision):**July 22, 2025**

Thanks for the additional attention to our MS. Responses to the review are below. Line numbers refer to the tracked MS.

In addition to the changes noted below in response to reviewers, we made many formatting and other changes to satisfy requests in the author checklist, all as discussed on the checklist that we have submitted with this revision.

Reviewer #1 (Remarks to the Author):

The comments/queries I had raised previously have been satisfactorily addressed. I only had a couple of additional minor comments for the authors' consideration:

1. Supplementary table 7 / pfmdr1 500N: since only pure mutants or pure wild-types were considered for these analyses, it would be interesting to add the results of the ex vivo RSA (less reliable in ex vivo assays with mixed infections as the authors reasonably posit) to the table. A similar trend, indicating an association between pfmdr1 500N and reduced DHA susceptibility, will presumably be observed. If not, it may be useful to add this point to the discussion.

RESPONSE: Interest in RSA results is appropriate, but as noted earlier in the Results section (line 171: "Interestingly, these mutations were not associated with RSA results.") and discussed in detail in the Discussion section (lines 215-226), our ex vivo RSA results did not correlate with identified resistance determinants. Adding RSA results to Supplementary Table 7 is not practical, as only a subset of samples received the RSA, and inclusion of separate "n's" for each assay will make the table very busy, with little added benefit. Rather, in response to this request we added another sentence to address the additional potential resistance markers (line 189): "As with the PfK13 mutations, the PfCARL D611N and PfMDR1 Y500N mutations were not associated with RSA results. "

2. Association of pfk13 C469Y, A675V with reduced lumefantrine susceptibility: The authors have explicitly acknowledged that there is no direct evidence implicating mutations in pfk13 with lumefantrine susceptibility. What they hint at but do not make as explicit (perhaps deliberately, in order not to be overly speculative) is the possibility that parasites with these mutations in pfk13 have acquired other mutation(s) which confer reduced susceptibility to lumefantrine. This would at least partially explain why these two mutations are increasingly prevalent, similar to the KEL1/PLA1 lineage which was predominant in the Greater Mekong Subregion. The authors may want to emphasise this point in the discussion section.

RESPONSE: In response we have more explicitly made the point suggested by our reviewer, adding a clause to this sentence in the Discussion (lines 234-237): "Although there is no direct evidence that decreased lumefantrine susceptibility is mediated by PfK13 mutations, it is possible that these PfK13 mutant parasites have acquired additional mutations that influence susceptibility to lumefantrine."

Reviewer #2 (Remarks to the Author):

I am happy with the responses to the reviewers.

I agree with the paper being accepted.

RESPONSE: Thanks.

Reviewer #3 (Remarks to the Author):

This an excellent work profiling ex vivo susceptibility to multiple drugs in Uganda at large scale. The authors have clarified the methods details and incorporated further commentary on the (uncertain) clinical consequences of the observed changes in artemether-lumefantrine

susceptibility in the region.

I am fully satisfied with the thorough responses.

RESPONSE: Thanks.